# Prolonged Sitting in University Students: An Intra-Individual Study Exploring Physical Activity Value as a Deterrent

**DOI:** 10.3390/ijerph20031891

**Published:** 2023-01-19

**Authors:** Alex C. Garn, Kelly L. Simonton

**Affiliations:** 1School of Kinesiology, Louisiana State University, Baton Rouge, LA 70803, USA; 2Division of Kinesiology and Health, University of Wyoming, Laramie, WY 82071, USA

**Keywords:** sedentary behavior, motivation, latent growth models, health behavior change

## Abstract

University students are a subpopulation of young adults highly susceptible to prolonged bouts of sitting. The purpose of this study was to investigate university students’ intra-individual patterns of prolonged daily sitting, exploring gender and physical activity value beliefs as covariates. A total of 71 students reported the number of days each week they spent in bouts of prolonged sitting (2 + continuous hours) over a five-week timeframe. The findings revealed that at the beginning of the study, the students spent about four days per week in prolonged bouts of sitting although there was substantial variability in the sample. Intra-individual changes over the five weeks occurred in a non-linear fashion with a variability in these trajectories. Men reported approximately one less day of prolonged sitting per week although gender did not predict changes over time. Physical activity value beliefs were negatively related to prolonged bouts of sitting when averaged across time. The results illustrate the variable nature of prolonged sitting in university student populations, highlighting the need for implementing individualized intervention strategies targeting sedentary behavior.

## 1. Introduction

Tremblay et al. define sedentary behavior as waking activities that involve reclined or sitting positions and low levels of energy expenditure [1]. Epidemiology researchers estimate that American adults spend over half their waking hours in sedentary behavior [2]. Researchers are starting to delve into different patterns of sedentary behavior [3] as data continuously show that prolonged amounts of continuous time being sedentary accompanies a myriad of health risks for non-communicable diseases and mortality [4,5,6]. There is an extensive body of research highlighting the benefits of physical activity for all segments of the population, including brain and heart health, bone strength, weight management, sleep quality, and enhanced health-related fitness [7].

A prolonged period of continuous sitting is one aspect of sedentary behavior receiving greater attention [6]. Specifically, researchers are starting to show that individuals who engage in prolonged periods of sitting experience increased health risks compared with peers with similar amounts of sedentary time without prolonged bouts of sitting [8]. However, this area of research is just starting to evolve so there are numerous aspects of prolonged sitting that have yet to be uncovered. Young adults enrolled in institutions of higher education represent a subgroup of the population identified as highly sedentary [9]. Contextual factors of higher education settings may promote prolonged periods of sitting [10]. For example, attending class, studying, and screen time are common activities for university students that facilitate prolonged periods of sitting. In a meta-analysis study by Castro et al., findings across 125 studies suggested that computer use was a prevalent sedentary behavior in university students that produced long periods of sitting [10]. Furthermore, homework demands often vary on a daily basis, making university students susceptible to prolonged sitting. Unpredictable schedules may also make it difficult to plan routine physical activity sessions and prevent habit formation [11]. Many young adults who attend universities live independently for the first time and experience greater levels of autonomy in their day-to-day decision-making, which can increase time spent in sedentary behavior and reduce engagement with health behavior [12]. Research has highlighted numerous benefits for university students who meet physical activity guidelines, including increased reports of quality of life and psychological wellbeing, as well as eating a healthy diet and acquiring adequate amounts of sleep [13].

We explored prolonged bouts of sitting in a sample of university students from an intra-individual perspective in this study. Most studies to date focus on inter-individual differences in sedentary behavior or prolonged sitting at a single timepoint [14], over time [15], or after an intervention [16]. A major gap in the current literature on university students’ prolonged sitting is research on intra-individual variability. Whereas inter-individual differences focus on trait-like averages, intra-individual variability relates to a state-like variability in those averages [17]. Exploring intra-individual variability allows researchers to examine dynamic changes in prolonged sitting in important ways. At the most basic level, intra-individual research can highlight how individuals’ prolonged sitting occurs. For example, it can show if prolonged sitting is generally stable for university students or if it changes in linear or non-linear patterns [18,19]. It can also underscore if changes in prolonged sitting occur in similar ways for different individuals. Another important element of intra-individual research is exploring factors that predict these changes in order to gain a comprehensive understanding of how to effectively intervene in university students’ prolonged sitting. In this study, we explored how the perceived values for physical activity and gender related to intra-individual variability in university students’ prolonged sitting patterns.

Researchers often cite a lack motivation for physical activity as a major antecedent of sedentary behavior in university students [19]. Value beliefs are one component of motivation that prominently contribute to behavioral decision-making [20,21,22,23]. Although numerous definitions exist for the value, in this study we operationalized the value as the relative worth that university students associated with physical activity [24]. Specifically, this reflected personal evaluations of the importance and usefulness of, as well as the interest in, physical activity. These three characteristics of value beliefs represent critical incentives for engaging in physical activity and potentially avoiding prolonged sitting as the value may prompt awareness to recognize when sedentary time has become excessive. For example, university students may break up long bouts of prolonged sitting because they believe the mental health benefits of physical activity are important, the weight management aspects of physical activity are useful for improving social opportunities, or simply because they find physical activity to be fun. Value beliefs can help provide an impetus for initiating physical activity engagement as well as increase intentions and satisfaction [25].

Previous research with university students has shown a positive correlation between the physical activity value and physical activity participation in the United States [22], Greece [26], and China [27]. Qualitative research has also revealed that university students often rely on value beliefs to make decisions about physical activity participation [20]. However, recent longitudinal research suggests that university students’ value beliefs may be more closely aligned with physical activity enjoyment than physical activity behavior [28]. Fewer studies have examined the interconnections between the physical activity value and sedentary behavior. Epstein et al. suggested that value beliefs play an important role in making choices about spending time in sedentary or physical activity behavior [29]. No studies that we are aware of have explored how the physical activity value beliefs relate to sedentary behavior (such as prolonged sitting) from an intra-individual perspective. In other words, there is currently no evidence relating to university students’ individual patterns and state-like dynamics of prolonged sitting.

Public health research consistently shows that, on average, males are more physically active than females [30], including in university student populations [31,32]. Researchers have hypothesized that gender differences in physical activity and sedentary behavior reflect a complex set of individual, social, and cultural factors [33,34]. Gender differences may be especially prevalent in higher education settings where females tend to spend more time in activities associated with sedentary behavior such as studying [35]. Intra-individual investigations of gender differences in prolonged sitting can provide insights into whether or not males and females change in similar or different ways. This type of evidence can advance our understanding of gender inequity patterns in university students and assist in the development of individualized intervention strategies.

The purpose of this study was to examine daily bouts of prolonged continuous sitting in university students, exploring the physical activity value as a potential deterrent. First, we investigated the intra-individual patterns of prolonged daily sitting across a five-week timeframe. Second, we evaluated how the intra-individual patterns of prolonged daily sitting varied between males and females. Finally, we explored the physical activity value as a time-varying covariate in order to understand its potential as a buffer to bouts of prolonged daily sitting. This was part of a larger study that focused on university students’ intra-individual physical activity patterns [28].

## 2. Materials and Methods

### 2.1. Participants

University students (*N* = 71) from a large university in the Southeastern United States participated in the study. The average age of the students was 21.25 (SD = 1.18) with slightly more males (55%) than females (45%). Most of the students reported their ethnic background as either White/Caucasian (66%), Black/African-American (21%), or Multi-Racial (4%). The academic rank of the students included seniors (i.e., year 4; 82%), juniors (i.e., year 3; 10%), and sophomores (i.e., year 2; 8%). All students in this convenience sample were majoring in kinesiology and had a good academic standing (i.e., a minimum grade-point average of 2.5 on a 4.0 scale). In order to be admitted into the kinesiology major, students needed a minimum of 24 earned credit hours and a cumulative grade-point average of 2.5. Participants, on average, reported engaging in approximately 120 min of moderate physical activity each week during the five-week study.

### 2.2. Procedures

The Institutional Review Board of the lead researcher granted permission to conduct the study (i.e., ethics approval was granted). We received permission from the instructor of a kinesiology course to visit their class and explain the study to the students. Afterward, the students received an email with an online survey link at the beginning of each week for five consecutive weeks. Participants provided informed consent to participate in the study before starting the first survey. Each survey link was active for 48 h in order to ensure responses occurred. This process created five waves of data with one-week intervals.

### 2.3. Measures

The online survey consisted of questions related to basic demographic characteristics, the physical activity value, and prolonged bouts of sitting behavior. The students answered the following three items about their physical activity value on a 7-point Likert scale ranging from strongly disagree (1) to strongly agree (7): (a) “Exercise is important to me”; (b) “Exercise is useful to me”; and (c) “Exercise is interesting to me”. Previous studies exploring the value in physical activity settings with university students have used similar items [22,27]. The students answered the following question related to prolonged sitting: “In the past week, on how many days did you spend sitting for a prolonged period of time (e.g., 2 h) with minimal amounts of movement?”. The answers ranged from 0 days to 7 days. This item was based on the 24-h movement guidelines [36].

### 2.4. Data Analysis

The preliminary analyses included calculations of the descriptive statistics and correlation estimates. The item-level reliability was measured for the three-item value scale using a coefficient alpha. However, it was not possible to use the coefficient alpha for the single-item scales because the within-person true score and error variances were indistinguishable. Therefore, we calculated a two-way random effects intra-class correlation (ICC2) on the prolonged sitting scores in order to demonstrate the average true score variance for individuals across the five timepoints. The main analyses consisted of a series of latent growth models with maximum likelihood estimation procedures using Mplus version 7.4. Full information maximum likelihood procedures (FIML) were used to handle the missing data [35]. We started with an intercept-only latent growth model when testing the intra-individual trajectories of prolonged sitting behavior whereby the intercept was freely estimated and the slope was constrained to zero (i.e., no growth). This was followed by testing models with a linear slope model and a linear and quadratic slope model, respectively. Finally, we tested a latent slope model that predicted the slope values for time two (T2) to T5. The residual variance estimates were constrained to be equal across the five timepoints.

We used joint criteria to evaluate the model fit for the latent growth models [37,38]. Specifically, we examined the chi-squared values based on the degrees of freedom, comparative fit index (CFI), Tucker–Lewis index (TLI), root mean square error of approximation (RMSEA), and standardized root mean residual (SRMR). Higher CFI and TLI scores (≥0.90 = adequate fit; ≥0.95 = good fit) and lower RMSEA and SRMR scores (≤0.08 = adequate fit; ≤0.06 = good fit) indicated better fitting models.

Once the best fitting growth model was determined for the intra-individual trajectories of prolonged sitting, we tested a prediction model that added gender as a time-invariant predictor and physical activity value scores as a time-varying predictor. Specifically, the model intercept and slope(s) for prolonged sitting were regressed on gender whilst the time-specific values of prolonged sitting were regressed on the time-specific values of the physical activity value (e.g., T1 sitting on the T1 value and T2 sitting on the T2 value, etc.). In the first prediction model, we freely estimated each time-specific regression between prolonged sitting and the physical activity value. In the second model, we constrained these time-varying regressions to be equal in order to obtain a standardized relationship between prolonged sitting and the physical activity value across the five waves of data.

## 3. Results

### 3.1. Descriptive Statistics

Missing data were relatively low across the five waves (T1 = 1%; T2 = 7%; T3 = 8%; T4 = 6%; T5 = 14%) and were handled with FIML. Table 1 provides an overview of the mean scores and standard deviations for the days of prolonged sitting per week and physical activity value, including a breakdown between males and females. Across the five-week period, the participants spent approximately 4 days per week with prolonged bouts of sitting (i.e., 2 h +). The lowest number of daily bouts occurred in week one (3.99; SD = 2.05) and the highest number of daily bouts occurred in week five (4.25; SD = 2.02). Females were more likely to report a higher number of daily bouts of prolonged sitting compared with males. The average physical activity value scores were stable across weeks one to four, with a slight increase in week five. These scores were well above the midpoint of the seven-point scale. The coefficient alpha estimates for the physical activity value across the five weeks ranged from 0.83 (week four) to 0.89 (week one). The ICC2 for prolonged sitting was 0.94, demonstrating a robust reliability. Table 2 highlights the bivariate correlations between prolonged sitting and the physical activity value. In general, the correlations showed weak-to-moderate negative relationships across the five waves of data.

### 3.2. Latent Growth Models

Table 3 reports the model fit estimates for all latent growth models. The findings revealed that whilst all growth models adequately fitted these data, the quadratic slope model produced the best results for representing the changes in prolonged sitting. Table 4 highlights the parameter estimates for the quadratic slope model. The latent intercept mean showed that the average number of prolonged sitting bouts at T1 was approximately four days per week with a variability around the mean. In other words, the days of prolonged sitting at the beginning of the study were not similar for the university students in this sample. Both the positive linear slope estimate and negative quadratic slope estimate were not statistically significant although both had a statistically significant variability. This suggested that the students’ daily bouts of prolonged sitting did not change at the same rates across the five-week study. Stated differently, the changes in the number of days these students spent in prolonged sitting were unique to individual students. The correlation between the intercept and slopes was not statistically significant, meaning that the rates of change were unrelated to the number of days the students reported in prolonged sitting at T1. Finally, there was a negative small-to-moderate relationship between the linear and quadratic slopes.

The model fit estimates for the prediction models are presented in Table 3 and the regression estimates are provided in Table 4. Both the full prediction model and standardized full prediction model produced a good model fit although the SRMR values were elevated. The findings showed that males reported approximately 1 day less per week of prolonged sitting at T1 (i.e., latent intercept). There was no relationship, however, between gender and the linear or quadratic slopes, showing that the rates of change were not associated with gender. Finally, the physical activity value was negatively associated with prolonged sitting at T1 and T2 in the full prediction model where the regression estimates varied at each timepoint. Overall, in the standardized model, the students’ average level of their physical activity value was negatively related to their average level of prolonged sitting across the five-week period.

## 4. Discussion

This study investigated university students’ weekly trajectories of prolonged sitting from an intra-individual perspective. The findings provide initial evidence on the state-like variability patterns associated with prolonged sitting across a five-week span. Furthermore, we explored individual differences in the intra-individual trajectories of university students’ prolonged sitting based on gender (i.e., time-invariant) and physical activity value beliefs (i.e., time-varying). Previous research has shown that high proportions of university students are sedentary [9] and do not meet physical activity guidelines [31]. However, the short-term stability and change in prolonged sitting in young adults attending university are relatively unknown. This study addressed that gap.

The results demonstrated that weekly changes in university students’ prolonged sitting in this sample followed a non-linear pattern, with individuals starting at different levels and showing different rates of change across time. Furthermore, the amount of prolonged sitting that these university students reported at the beginning of the study was unrelated to the rate of change across the five weeks. During an average week, these students reported spending about four days with at least one prolonged bout of sitting. The findings provide further evidence that many university students spend extensive amounts of time in sedentary behavior [9,12]. Similarly, the prolonged sitting behavior of these students contrasted with the 24-h movement recommendations [36]. This adds to the accumulating evidence suggesting that higher education contexts facilitate health barriers that need direct interventions [39].

Finding effective ways to target prolonged bouts of sedentary behavior is an area of research gaining momentum [10,40]. The significant variability in the latent mean and latent slopes of the growth models suggests that prolonged sitting behavior is different for individual students, including the number of days spent in extended bouts of sitting and the rate of change across weeks. Therefore, any single standardized approach is unlikely to provide a comprehensive solution. Taking an individualized approach would likely be more effective. Previous research has indicated that many university students are not concerned about [40] or unaware of [10] the health consequences of prolonged bouts of sedentary behavior. Therefore, increasing students’ knowledge about the health risks associated with prolonged bouts of sitting via information strategies may be a starting point for developing individualized intervention strategies.

Castro et al. outlined the importance of using diverse behavior change strategies in order to maximize the impact on university students’ prolonged sedentary behavior [10]. Five general areas, including informational, behavioral, social, psychological, and motivational, can be used to tailor intervention strategies. Informational examples include developing campus-wide email campaigns or hanging fact-based posters in classrooms. Behavioral examples include instituting activity breaks in classes as well as setting personal goals and developing action plans related to breaking up sedentary behavior. Social examples include developing physical activity social networks and social support groups at different university levels (e.g., major/department, college, and campus). Psychological examples include teaching students self-monitoring/self-regulation skills. University systems could develop holistic approaches to reduce sedentary behavior such as prolonged sitting that targets individual needs, values, and contextual circumstances.

Previous research has demonstrated that the physical activity value is closely associated with increased physical activity [20,22,27]. Our results add to this literature by providing an insight into using strategies related to students’ physical activity motivation to minimize sedentary behavior. Specifically, finding ways to increase the university students’ physical activity value appears to have merit in potentially reducing prolonged bouts of sitting. Previous research in the academic domain shows that interventions can be successful at increasing university students’ value beliefs [41]. Intervention strategies could focus on having students actively reflect and create specific examples of how learning academic content such as math and science is relevant and meaningful in life outside the classroom. We suggest that same general approach could be used to increase physical activity and reduce prolonged sitting. In addition, other strategies such as using existing technologies (e.g., smart watches and apps) to provide supportive reminders that are tailored to student schedules may be one example of utilizing systems in place to create awareness, along with building a perceived value for activity.

Our findings suggest that these strategies need to be dynamic and delivered in short-term intervals in order to match the university students’ patterns of behavior. Similarly, implementing strategies based on gender does not appear to be an effective strategy either. In other words, university students’ prolonged sitting patterns were highly variable in this sample, suggesting the need to create interventions that utilize individualized approaches. Motivational interviewing is one intervention strategy that seems to be well-suited to the diverse patterns highlighted in this study. Motivational interviewing is a client-centered therapeutic strategy that focused on targeting individual facilitators and barriers to enhance health behavior changes specific to the person [42]. Delivering motivational interviewing services in higher education contexts could occur using diverse mechanisms such as personal wellness classes, health center services, or wellness fairs.

### Limitations and Future Research

The current study had a few limitations. Although there were over 300 observations analyzed in this study, the overall sample size was small, which should be considered when interpreting the results. It should also be noted that all students were from a single major (i.e., kinesiology) that was focused on physical activity. Replication with larger samples of university students in future research would advance the generalizability. It is also important to note that the question related to bouts of prolonged sitting had a reference time of 2 h. This could be considered severe because many public health researchers suggest as little as 30 min of prolonged sitting can produce health problems [10,43]. Future research would benefit from investigating a prolonged sitting time in shorter intervals. Additionally, this study did not include data on contextual factors such as the number of course credit hours, frequency of class meetings, and other school-related (e.g., labs and online course requirements) and non-school-related demands (e.g., employment and social relationships). Future research should also use measure body composition indices and use device-based measures such as accelerometers in order to provide a more objective measure of prolonged sitting that is better suited for estimating energy expenditure.

All university students in this sample were from the same region of the United States, which could affect the generalizability of the results. Similarly, approximately 80% of the sample reported their ethnic background as White/Caucasian. Future research should randomly sample a more diverse group of university students. The value beliefs in this study focused on physical activity; however, it would be prudent for future research to examine the value beliefs for sedentary behavior as well. In fact, it may be beneficial to explore the differentials between the physical activity value beliefs and sedentary value beliefs to obtain a full picture of the relationship between the value beliefs and prolonged sitting. Finally, this study focused on short-term (i.e., weekly) bouts of prolonged sitting, which may not reflect the stability and changes over longer periods. Future research should explore longer intervals of time in order to gather a more comprehensive understanding of university students’ patterns of prolonged sitting.

## 5. Conclusions

The individual patterns of prolonged sitting were quite variable in this sample of university students. Although males reported spending about one less day with a prolonged bout of sitting at the beginning of the study, gender did not explain the variability in individual patterns. The physical activity value was negatively related to prolonged sitting during the first two weeks and overall, when averaged across the five weeks. However, the value beliefs did not predict prolonged sitting in the final three weeks of the study. Taken together, the findings showed that university students’ prolonged sitting in the context of this study was a highly individualized behavior. This is important to understand when considering how to address prolonged sitting effectively in this population. Our results suggested that a one-size-fits-all approach was unlikely to work for students attending this particular university. Therefore, creating individualized behavior change strategies appears to be necessary to enhance health interventions aimed at reducing this type of sedentary behavior.

## Figures and Tables

**Table 1 ijerph-20-01891-t001:** Descriptive statistics of prolonged sitting and physical activity value.

	Total		Males		Females	
	M	SD	M	SD	M	SD
Week 1 Sitting	3.99	2.05	3.39	2.14	4.69	1.73
Week 2 Sitting	4.06	1.97	3.85	2.12	4.28	1.80
Week 3 Sitting	4.20	2.12	3.91	2.21	4.50	1.98
Week 4 Sitting	4.16	2.16	3.69	2.17	4.69	2.06
Week 5 Sitting	4.25	2.02	3.94	2.05	4.61	1.97
Week 1 Value	5.50	1.35	5.73	1.26	5.24	1.43
Week 2 Value	5.53	1.20	5.76	1.05	5.28	1.31
Week 3 Value	5.56	1.19	5.83	1.10	5.29	1.22
Week 4 Value	5.50	1.22	5.70	1.15	5.28	1.27
Week 5 Value	5.72	1.18	5.91	1.07	5.05	1.27

Sitting refers to days per week of 2 + hours of prolonged sitting. Value refers to physical activity value beliefs.

**Table 2 ijerph-20-01891-t002:** Correlation estimates of weekly prolonged sitting and physical activity value.

		1	2	3	4	5	6	7	8	9	10
1	Week 1 Sitting	1.00									
2	Week 2 Sitting	*0.63*	1.00								
3	Week 3 Sitting	*0.70*	*0.82*	1.00							
4	Week 4 Sitting	*0.63*	*0.80*	*0.80*	1.00						
5	Week 5 Sitting	*0.63*	*0.77*	*0.72*	*0.82*	1.00					
6	Week 1 Value	*−0.37*	−0.19	*−0.25*	−0.23	−0.14	1.00				
7	Week 2 Value	*−0.37*	−0.18	*−0.28*	−0.24	−0.15	*0.78*	1.00			
8	Week 3 Value	*−0.39*	*−0.31*	*−0.31*	*−0.36*	−0.23	*0.81*	*0.86*	1.00		
9	Week 4 Value	*−0.31*	−0.17	*−0.26*	−0.22	−0.15	*0.71*	*0.83*	*0.80*	1.00	
10	Week 5 Value	*−0.34*	−0.14	*−0.25*	−0.13	−0.20	*0.67*	*0.82*	*0.70*	*0.87*	1.00

Values in italics indicate statistical significance; *p* < 0.05.

**Table 3 ijerph-20-01891-t003:** Model fit estimates of latent growth model analyses.

Model	χ²	*df*	*p*	CFI	TLI	RMSEA	SRMR
Sitting							
Intercept-Only	28.55	17	0.03	0.96	0.97	0.098	0.072
Linear	20.78	14	0.11	0.97	0.98	0.083	0.072
Quadratic	9.61	10	0.48	1.00	1.00	0.011	0.037
Latent	11.40	10	0.47	0.99	0.99	0.044	0.042
Sitting with Predictors							
Full Model	47.28	42	0.26	0.98	0.98	0.042	0.198
Standardized Full Model	49.08	46	0.35	0.99	0.99	0.031	0.192

CFI: comparative fit index; TLI: Tucker–Lewis index; RMSEA: root mean square error of approximation; SRMR: standardized root mean residual.

**Table 4 ijerph-20-01891-t004:** Parameter estimates from latent growth model analyses.

Parameter	I Mean	I Variance	S Mean	S Variance	S2 Mean	S2 Variance	I S Cov	I S2 Cov	S S2 Cov
Sitting	3.95 (0.23) **	3.10 (0.65) **	0.08 (0.18)	1.12 (0.42) **	−0.01 (0.04)	0.05 (0.02) *	−0.35 (0.38)	0.06 (0.09)	−0.26 (0.09) *
Covariates									
I on Male	−1.01 (0.43) *								
S on Male	0.44 (0.35)								
S2 on Male	−1.02 (0.08)								
Sit_1 on Val_1	−0.38 (0.13) **								
Sit_2 on Val_2	−0.25 (0.12) *								
Sit_3 on Val_3	−0.18 (0.14)								
Sit_4 on Val_4	−0.22 (0.14)								
Sit_5 on Val_5	−0.29 (0.16)								
Sit_Ave on Val_Ave	−0.29 (0.10) *								

I: latent intercept; S: latent slope; Cov: covariance; Sit_1: prolonged sitting at time one; Val_1: physical activity value at time one. * *p* ≤ 0.05, ** *p* ≤ 0.01.

## Data Availability

Data is available upon request to the first author.

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
