# Peer review of "Prolonged Sitting in University Students: An Intra-Individual Study Exploring Physical Activity Value as a Deterrent"

_ijerph, 2023, doi:10.3390/ijerph20031891_

Round 1

Reviewer 1 Report

The introduction responds to the objectives set out in the research. The authors present a very complete introduction with a high percentage of references from the last five years. However, I believe it is important to be able to provide studies on the possible differences that may exist in the levels of physical activity depending on the university studies selected in the sample.

With reference to the section on materials and methods, a correct description of the selected sample is given. However, the procedure section needs to be developed in more depth. Similarly, the online survey conducted should present some internal consistency test, such as Cronbach's Alpha, for example. The statistical analysis is in line with the objectives of the study and presents a complete design. The tables presented by the authors facilitate the interpretation of the data.

In reference to the discussion, I again refer to the need to take into account the type of university studies in order to be able to make a comparative assessment of university students and physical activity. There is also no reference to the lower values of physical activity in women compared to men, why? I think the authors make a good self-critical analysis of their limitations and propose new avenues for future work.

1. As I stated in my initial review, the authors do not take into account the differences that may exist between the different curricula of university students in relation to sedentary periods. It should be noted that there are numerous scientific evidences and articles that take into account these factors and the measurement of the metabolic rate (MET) of the intensity of physical activity performed by the subjects. For example, the sedentary time of a communications engineering student is far from the sedentary time of a student of physical activity and sports science. Likewise, the degree of difficulty of the studies themselves (which can be estimated by the entrance marks for these studies), can mark differences in the values of sedentary time among university students.

2. Neither has the body composition of the sample been taken into account, an aspect which, once again, scientific evidence considers to be fundamental for energy expenditure at rest and to be taken into account in times of physical inactivity.

3. With reference to the procedure and the instrument used in the research by the authors, I have some questions: is it a standardised test? If not, how was it designed? did it have a testing and validation phase? was there any kind of test to assess the internal consistency of the instrument? why was it only active for 48 hours and why, being an online instrument, were other university students from other types of studies not invited to participate? What was the purpose of the five waves of data obtained if there is no intervention with the sample during these periods? With reference to the responses, were they likely to create profiles of the participants in the sample? If there was such a risk, has the research been endorsed by an Ethics Committee?

4. In relation to the female sample. There is sufficient scientific evidence that the abandonment of regular physical activity with the onset of adolescence is greater in women than in men. Has this aspect been taken into account in obtaining the results and subsequent conclusions? There is no mention of this in the paper and it may constitute a bias in the research. Furthermore, it should be noted that although the sample of men and women is balanced, 55% and 45% respectively, the sample is not small, which constitutes a further limitation for the generalisation of the results, especially in women, who do not have a specific treatment. In my view, this should be reflected in the manuscript.

I hope that these more concrete comments will help to determine the necessary changes in the authors' work. 

Author Response

The introduction responds to the objectives set out in the research. The authors present a very complete introduction with a high percentage of references from the last five years. However, I believe it is important to be able to provide studies on the possible differences that may exist in the levels of physical activity depending on the university studies selected in the sample.

With reference to the section on materials and methods, a correct description of the selected sample is given. However, the procedure section needs to be developed in more depth. Similarly, the online survey conducted should present some internal consistency test, such as Cronbach's Alpha, for example. The statistical analysis is in line with the objectives of the study and presents a complete design. The tables presented by the authors facilitate the interpretation of the data.

In reference to the discussion, I again refer to the need to take into account the type of university studies in order to be able to make a comparative assessment of university students and physical activity. There is also no reference to the lower values of physical activity in women compared to men, why? I think the authors make a good self-critical analysis of their limitations and propose new avenues for future work.

  1. As I stated in my initial review, the authors do not take into account the differences that may exist between the different curricula of university students in relation to sedentary periods. It should be noted that there are numerous scientific evidences and articles that take into account these factors and the measurement of the metabolic rate (MET) of the intensity of physical activity performed by the subjects. For example, the sedentary time of a communications engineering student is far from the sedentary time of a student of physical activity and sports science. Likewise, the degree of difficulty of the studies themselves (which can be estimated by the entrance marks for these studies), can mark differences in the values of sedentary time among university students.

Response: Thank you for reviewing our manuscript and providing meaningful feedback. We had added more information about the sample including the fact that all students were kinesiology majors. In the initial submission, we identified the lack of contextual information about the students as a limitation. See lines 314-317. We also added information on the standards to get into this major (e.g., entrance marks). See lines 122-127.  We have also added this as a limitation. See lines 308-309.

  1. Neither has the body composition of the sample been taken into account, an aspect which, once again, scientific evidence considers to be fundamental for energy expenditure at rest and to be taken into account in times of physical inactivity.

Response: Prolonged sitting was measured as a behavioral variable. We did not measure energy expenditure. In the future research section we have added the need for future researchers to explore aspects of energy expenditure associated with prolonged sitting. See lines 318-320.

  1. With reference to the procedure and the instrument used in the research by the authors, I have some questions: is it a standardised test? If not, how was it designed? did it have a testing and validation phase? was there any kind of test to assess the internal consistency of the instrument? why was it only active for 48 hours and why, being an online instrument, were other university students from other types of studies not invited to participate? What was the purpose of the five waves of data obtained if there is no intervention with the sample during these periods? With reference to the responses, were they likely to create profiles of the participants in the sample? If there was such a risk, has the research been endorsed by an Ethics Committee?

Response: The online survey was made up of scales and questions patterned after previous research exploring physical activity value and prolonged sitting. We identify studies that the physical activity value scaled was patterned after (see line 143 ) and the prolonged sitting question was based on the 24-hour movement guidelines (see line 147). We report coefficient alpha scores for the value scale on lines 192-193, which is possible because it has multiple items. However, prolonged sitting was measured with a single item so coefficient alpha is not appropriate. We have added an intra-class coefficient estimate for prolonged sitting in order to demonstrate its robust reliability across time– see lines 193-194 . We have added that this was a convenience sample – see line 122. We created a 48 hour window so that participants would report their behavior in a timely manner reducing lags in reporting that could negatively influence memory. The rationale of the five-week timeline without an intervention was to investigate the within-person variability of university students’ prolonged sitting in their natural setting without intervention. Yes the research was endorsed by lead researcher’s Internal Review Board (i.e., Ethics approval) – see lines 129-130.

  1. In relation to the female sample. There is sufficient scientific evidence that the abandonment of regular physical activity with the onset of adolescence is greater in women than in men. Has this aspect been taken into account in obtaining the results and subsequent conclusions? There is no mention of this in the paper and it may constitute a bias in the research. Furthermore, it should be noted that although the sample of men and women is balanced, 55% and 45% respectively, the sample is not small, which constitutes a further limitation for the generalisation of the results, especially in women, who do not have a specific treatment. In my view, this should be reflected in the manuscript.

Response: We agree with your comments, which is why we included a paragraph in the original introduction on gender (see lines 97-106), gender (i.e., males coded as 1) was used as a predictor in our final model (see Table 4). We also included a paragraph in our discussion on gender (see lines 294-304).

I hope that these more concrete comments will help to determine the necessary changes in the authors' work.

Response: Thank you for your feedback, we appreciate your thoughtful comments.

Reviewer 2 Report

Dear Authors,

The work is interesting and relevant.

Regardless, I have a few remarks that could improve this article.

The introductory section is severely lacking in insights that reveal the specific benefits of physical activity for the student/young adult.

In the section Materials and Methods it would be good for readers to see participants study programs, as it can lead to their sitting behavior. Also, if it is possible, it would be great for the readers to know what  the physical activity experience of the subjects was at school and in the family.

Also, the position of the authors to generalize the results of the study, obtained after examining 71 subjects, to the entire student population is also controversial. I would suggest making the research conclusion more specific. At this point, it would be very useful to distinguish the field of study.

Author Response

Dear Authors,

The work is interesting and relevant.

Regardless, I have a few remarks that could improve this article.

The introductory section is severely lacking in insights that reveal the specific benefits of physical activity for the student/young adult.

Response: Thank you for highlighting this oversight. We have added information about the benefits of physical activity to the revised introduction.  See lines 29-31 and 49-52.

In the section Materials and Methods it would be good for readers to see participants study programs, as it can lead to their sitting behavior. Also, if it is possible, it would be great for the readers to know what  the physical activity experience of the subjects was at school and in the family.

Response: We have added additional information related to the participants’ study program as well as their self-reported physical activity during the five-week study. See lines 122-127.

Also, the position of the authors to generalize the results of the study, obtained after examining 71 subjects, to the entire student population is also controversial. I would suggest making the research conclusion more specific. At this point, it would be very useful to distinguish the field of study.

Response: Thank you for pointing this out. We changed language throughout the Discussion and Conclusion sections to focus more on generalizations to university students in this study context. See lines 247, 298, 334, 340-343.

Round 2

Reviewer 2 Report

Dear authors, Your corrections really bring clarity to the research.